# Effect of Variation of SiC Reinforcement on Wear Behaviour of AZ91 Alloy Composites

**DOI:** 10.3390/ma14040990

**Published:** 2021-02-19

**Authors:** Anil Kumar, Santosh Kumar, Nilay Krishna Mukhopadhyay, Anshul Yadav, Virendra Kumar, Jerzy Winczek

**Affiliations:** 1Department of Mechanical Engineering, Kamla Nehru Institute of Technology, Sultanpur 228118, India; anilk@knit.ac.in (A.K.); veer.iitdmech@gmail.com (V.K.); 2Department of Mechanical Engineering, Indian Institute of Technology (BHU), Varanasi 221005, India; santosh.kumar.mec@itbhu.ac.in; 3Department of Metallurgical Engineering, Indian Institute of Technology (BHU), Varanasi 221005, India; mukho.met@iitbhu.ac.in; 4Membrane Science and Separation Technology Division, CSIR-Central Salt and Marine Chemicals Research Institute, Bhavnagar 364002, India; anshuly@csmcri.res.in; 5Department of Technology and Automation, Częstochowa University of Technology, 42-201 Częstochowa, Poland

**Keywords:** wear, metal matrix composites, SiC, AZ91, magnesium alloy

## Abstract

In this investigation, the extensive wear behaviour of materials was studied using SiC reinforced magnesium alloy composites fabricated through the stir casting process. The wear properties of AZ91 alloy composites with a small variation (i.e., 3%, 6%, 9% and 12%) of SiC particulates were evaluated by varying the normal load with sliding velocity and sliding distance. The worn surfaces were examined by scanning electron microscope to predict the different wear mechanisms on the pin while sliding on the hard disk in the dry sliding wear test condition. The microhardness of the SiC reinforced AZ91 composites was found to be more than the un-reinforced AZ91 alloy. Pins tested at load 19.62 N, and 2.6 m/s exhibited a series of short cracks nearly perpendicular to the sliding direction. At higher speed and load, the oxidation and delamination were observed to be fully converted into adhesion wear. Abrasion, oxidation, and delamination wear mechanisms were generally dominant in lower sliding velocity and lower load region, while adhesion and thermal softening/melting were dominant in higher sliding velocity and loads. The wear rate and coefficient of friction of the SiC reinforced composites were lower than that of the unreinforced alloy. This is due to the fact of higher hardness exhibited by the composites. The wear behaviour at the velocity of 1.39 m/s was dominated by oxidation and delamination wear, whereas at the velocity of 2.6 m/s the wear behaviour was dominated by abrasion and adhesion wear. It was also found that the plastic deformation and smearing occurred at higher load and sliding velocity.

## 1. Introduction

Composite materials are imperative for the design of materials because of their extraordinary mechanical properties. Composite materials in which discrete materials are joined mechanically can offer better properties than conventional monolithic materials, such as high hardness and wear resistance. Metal matrix composites (MMCs) have been developed to enhance mechanical properties, including specific strength, specific modulus, and good wear resistance. There has been an interest and intense research for developing composites that contain low density and low-cost reinforcements.

Magnesium-based alloys are getting attention due to their lightweight capabilities. They also exhibit good castability, weldability, machinability, thermal stability, specific mechanical properties, and resistance to electromagnetic radiations [1,2,3,4,5]. The limitation of magnesium-based materials arises from its low ductility, modulus, and poor corrosion resistance. AZ91 is one of the commercial magnesium alloys being used extensively in the automobile and aerospace industries. The metal matrix composites of AZ91 magnesium alloy with different types of ceramics and metallic reinforcements were fabricated. Their physical, mechanical properties were evaluated by various researchers [2,4,6,7] to know the effect of the addition of different reinforcements.

Several parts of an automobile such as the piston, cylinder bores, brakes of the engine, and the other components are exposed to sliding motion. The tribological behaviour is affected by multiple factors for the dry sliding system. It depends on test material properties, counterface material, environment and experimental parameters like sliding velocity, applied load, and sliding distance. Chen and Alpas [8] examined the dry sliding wear of AZ91 alloy versus AISI52100 steel as a counterface. Limited research has been done on tribological properties of the magnesium-based metal matrix composites. Sharma et al. [9] found that the Mg-alloy wear rates reinforced with feldspar particles decreased with increasing the amount of reinforcement. The effect of variation of load resulted in the transition of mild to severe wear. Thakur and Dhindaw [10] investigated that the distribution of SiC reinforcements in magnesium alloy influenced the tribological properties of the MMCs during sliding wear test. Aluminium alloy composites reinforced with SiC_p_ have been used mostly in commercial production [11]. These composites can offer superior tribological and mechanical properties at a relatively lower cost [12,13,14].

Wang et al. [15] studied the mechanical and wear properties along with microstructure of Grp/AZ91 magnesium matrix composites. The study found that the grain size of 5% Grp/AZ91 composite initially decreased and increased with the increase of forging pass. Patil et al. [16] studied the SiC/fly ash reinforcement effect on surface properties of Al7075 hybrid composites. The study found that the SiC/fly ash interaction vol % and hybrid ratio had affected the highest on the wear rates and microhardness of composites. The increase in microhardness was observed with an increase in a volume percentage of SiC/fly ash powders.

Mg and its alloys can replace iron and aluminium alloy in automobile and aerospace industries due to their lower density and higher specific strength [17,18]. However, its low ductility and resistance to corrosion and wear have been a serious issue preventing it from being used extensively as an Al–Cu-alloy [19,20,21,22]. Some studies have recently been conducted on the magnesium alloy matrix composite, which exhibits higher properties than monolithic alloys [23,24,25,26]. Magnesium alloy-based composites with intermittent reinforcement promise high specific strength and stiffness, dimensional stability, good damping capacities, and creep resistance. The various researchers developed magnesium-based composites reinforced with nano-alumina, and they reported that the abrasion and the adhesive wear were the common wear mechanisms [27,28]. Dry sliding wear performance of AZ91 composites reinforced with 5 and 10 vol% SiC_p_ and rheocast AZ91 alloy have also been investigated under different normal loads and sliding speeds by the dry sliding pin on disc arrangement against steel disc [1]. The ultimate tensile, yield and compression strength are increased with an increase in TiC reinforcement in AZ91. The sliding wear rate increases with an increase in normal load and sliding speed [3,4]. The benefits of magnesium and its alloy used as a composites matrix are the high specific strength and stiffness, dimensional stabilities, and good damping capacities [29]. The mechanical and tribological behaviour of the MMCs has been studied to know the effect of the reinforcement. Some information concerning the wear behaviour of magnesium-based MMCs reveals that magnesium alloys’ tribological properties can be improved with the addition of hard ceramic fibre or particulates reinforcements [27,30,31,32,33,34].

There is no information available concerning Mg alloy composites’ wear properties fabricated through vacuum-assisted inert atmosphere stir casting process with a small variation (3%, 6%, 9% and 12%) of SiC_p_. This study’s primary aim is to investigate the matrix alloy’s sliding wear behaviour and its composite with varying content of SiC_p_.

## 2. Material and Method

Various composites of AZ91 alloy were prepared by adding a different percentage of SiC_p_ of average size ~15 μm. The SEM image of SiC reinforcement is presented in Figure 1. The magnesium alloy was loaded in stir casting furnace first and then vacuum was created to remove oxygen present inside the crucible, die and holder. Preheated SiC_p_ with the help of anti-chamber were added after complete melting of alloy. The cast samples were cut and machined into appropriate sizes for further testing and characterisation. The major types of equipment used in this study were hardness tester (LECO’s LV Series, Geleen, Netherlands), ZEISS scanning electron microscope (Oberkochen, Germany) equipped with E.D.S. and Ducom pin on disc tribometer (Bangalore, India). Vickers microhardness testing was carried out to understand the effect of the distribution of SiC within magnesium alloy (AZ91). A rectangular cross-section sample (15 mm × 30 mm) of fabricated composites was used to measure the microhardness. The microhardness tests were conducted throughout the cross-section at ten different points (at the load of 9.81 N) of the AZ91 alloy and its composites reinforced with SiC_p_. Pin on disk friction and wear testing machine was used to evaluate the tribological characteristics and its composites. The wear sample was made as per ASTM standard: G99 [35]. The friction wear weight losses were generated by varying the normal load at two different velocities for all the composites. The wear measurement was reported as the volume loss in cubic millimetres for the pin. The Equation (1) is used to calculate wear volume loss, assuming that there is no significant wear on the disk.
(1)$$Pin\;volume\;loss\;\left({mm}^{3}\right)=\frac{Weight\;loss\;\left(g\right)}{Density\;\left(\frac{g}{cm^{3}}\right)}\times 1000$$

The density measurement was carried out in agreement with the Archimedes principle. Distilled water was taken as the immersion fluid. Three samples were taken in each case to weigh both in the air and when fully immersed in distilled water.

The specimens’ weight loss was calculated by evaluating weight before and after the test using an Ohaus make balance having an accuracy of 0.0001 g. The wear rates were calculated from the volume loss divided by sliding distance. The average of the three measurements of the wear tests was taken to observe the wear rate. The worn surfaces were cleaned thoroughly to remove the wear debris, and then it was observed using scanning electron microscopy (SEM) technique.

The dry sliding wear test for AZ91 alloy and its composite with the different percentage (3, 6, 9 and 12) of SiC_p_ was performed. The variations of sample weight loss with constant sliding speed were observed under variable normal load from 9.81 to 58.86 N. The tribological pairs were selected by varying the normal load at constant velocity (or distance). The different parameters used in dry sliding wear test are tabulated in Table 1. The average roughness (*R_a_*) of disc material and pin materials were 0.1 and 0.8 μm, respectively.

## 3. Results and Discussion

Figure 2 shows the average microhardness value of magnesium alloy (AZ91) and (3, 6, 9, 12) wt % of SiC_p_ reinforced composites. The variation in the microhardness value of magnesium alloy (AZ91) and its composites is due to the presence of hard particles in the soft matrix. The load-bearing capacity of composites is more than monolithic alloy due to the presence of reinforcement. The hardness value was increased by 15% on the addition of 3 wt % of SiC_p_, and it further increased up to 73% on the addition of 12 wt % of SiC_p_.

The wear weight loss of AZ91 alloys and SiC_p_ reinforced composite were plotted against different normal loads at the constant sliding speed of 1.39 m/s in Figure 3. The wear weight loss of unreinforced magnesium alloy was found to have a higher value than that of the composites at all the normal load for sliding speed of 1.39 m/s as seen in Figure 3. As the percentage of reinforcement increased in the composite, the wear weight loss decreased for all the different (3, 6, 9 and 12) percentage of the SiC_p_ reinforcements. However, weight loss for the composite increased with an increase in normal load. The estimation of wear weight or wear volume may be estimated by Archard’s law [36]. According to Archard’s law, material loss in wear is directly proportional to sliding distance and normal applied load but indirectly proportional to the material’s hardness. The Archard’s law is applicable mainly for single-phase materials, but it works fairly well for multiphase alloys and composites. This law has been proposed only for adhesion wear. Therefore, the composite materials are observed to be slightly less severe than monolithic alloys. The composite materials having higher hardness than unreinforced alloy due to hard SiC_p_ can exhibit less wear as per Archard’s law. The Archard law can be written as:(2)$$V=\frac{\;K_{1}Nl}{3H},$$

*V*—volume loss in wear, *N*—the normal applied load, *l*—the sliding distance, *H* —the test materials’ hardness. *K_1_* is a wear constant depends on elastic and plastic contacts, shearing of those contacts, mode of lubrication, the effect of environment, etc. The adhesive wear is generally associated with low normal load and low sliding velocity. The others wear mechanisms are also involved in the material removal process of dry sliding wear.

The combined effect of the different wear mechanism is involved in the estimation of wear loss. The addition of hard particle (SiC) in base alloy AZ91 may provide dry lubrication as well as rolling contact between the hard disk and soft pin materials. Therefore, on an increasing amount of reinforcement, the wear weight loss and wear volume decreased in the case of particulates reinforced composites. It seems that wear rate is greater at the higher load and the wear rate decreased on increasing the weight percentage of SiC_p_ reinforcement. The wear rate plot concerning applied normal load is almost linear, which is in conforming to the Archard’s law. The variations of the average sliding coefficient of friction (*µ*) with a different value of applied normal loads are shown in Figure 4. The average coefficient of friction decreased as the normal load increased in all the cases, irrespective of alloy and composite. The average coefficient of friction value at lower load was slightly different from its behaviours at higher loads. The unreinforced alloy had a higher value of the coefficient of friction than SiC_p_ reinforced composite. Researchers have reported a decrease in the average coefficient of friction with an increase in the normal applied load [37,38]. However, the extent of decrease was to be correlated with types, size, and distribution of reinforcements. It is clear from Figure 4 that the coefficient of friction for magnesium alloy and its composites decreased with an increase in normal load value. The coefficient of friction also decreased with an increase in the percentage of SiC reinforcement. The small variation in coefficient of friction was observed in composites at higher load and a higher percentage of reinforcements. This behaviour can be explained with the phenomenon that on increasing applied normal load, the mating surfaces’ asperities will be deformed, leading to a decrease in frictions’ coefficient.

The wear rate of magnesium alloy and composites is a function of sliding velocity, normal load, and environmental conditions. The variation of wear rate at different velocity and load is due to different wear mechanisms, which may be an extraordinarily complex phenomenon to understand. The wear rates at two different speeds concerning three different loads are presented in Figure 5a–e for the monolithic alloy and it composites. It can be concluded that the wear rates at the sliding speed of 2.60 m/s are higher than the wear rates at the sliding speed of 1.39 m/s at all applied load irrespective of base alloy and composites. The difference in wear rates at these two velocities (1.39 m/s and 2.60 m/s) was highest in case of base alloy AZ91. However, in the case of composites, this difference decreased as the percentage of reinforcement increased. The lowest difference was observed in 12% SiC reinforced composites. The purpose of these figures was to show the effect of velocity and load on the same alloy and composites. The wear rate at a velocity of less than 1 m/s was different from the wear rate at a velocity higher than 1 m/s, as reported by Lim et al. [13]. A similar trend was also observed in current research that increasing velocity or load the wear rate generally increased.

The optical micrograph of composites at a different percentage of SiC_p_ was shown in Figure 6. The distribution of SiC_p_ in AZ91 magnesium alloy seems to be reasonably uniform, but in some places, particles’ accumulations were observed. The primary magnesium dendrites (α phase) which are coexisting with dark inter dendrite precipitate of intermetallic β phase. The dark blue arrows in Figure 6 represents α and the β phases are represented by saffron arrows. The β phase generally has the composition of Mg_17_ (Al, Zn)_12_. The reinforced particles are represented by light blue arrows.

The worn pin surfaces were examined by scanning electron microscope to predict the different wear mechanism during sliding on the hard disk in dry sliding wear test. Figure 7 shows the selective SEM micrographs of worn surfaces of the magnesium alloy AZ91 at different load, velocity, and magnifications. Several grooves and scratches can be seen on almost all the worn pin within this velocity and load range. The oxidation wear was observed at low velocity lower normal load. Figure 8 shows the selective SEM micrograph of SiC reinforced AZ91 magnesium alloy composites at different load, velocity, and weight percentage of reinforcements. In the case of composite ploughing delamination and oxidation, wear was prominently present in the worn surfaces. A large amount of accumulated embedded debris was also observed at the worn surfaces of the pin. The energy-dispersive X-ray spectroscopy (Figure 9) identified more substantial oxygen peaks than Al and Zn, indicating the presence of magnesium oxide in the wear debris. This behaviour is suggestive of oxidative wear, which was developed due to frictional and sliding heating of the pin materials. Composite materials exhibited the extensive formation of oxide than unreinforced magnesium alloy. Five different wear mechanisms may be operated singly or in combination under different normal load and sliding speed. The exact quantification of any wear mechanism was not possible. It was also not possible to exclude any wear mechanism entirely out of these five. The wear mechanism may be in the combination of abrasion, oxidation, delaminating, adhesion, thermal softening and melting. It is challenging to examine the exact influence of one or others wear mechanism correctly. The wear behaviour within this velocity was oxidation at lower load and delamination or abrasion wear at higher load. Several grooves and scratch marks parallel to the sliding direction was signature of abrasion wear. The de-attachment of wear particle occurs in the form of sheet and short cracks roughly perpendicular to the sliding direction. This type of wear mechanism classified as delamination wear was first introduced by Suh [39,40]. The oxidation wear also occurred on the material surface under the test conditions, and due to the oxidation, it became dark at the wear surfaces.

A summary of different wear mechanisms for a different combination of sliding velocity and the applied normal load was compiled in Table 2. Pins tested at load 19.62 N, and 2.6 m/s exhibited a series of short cracks nearly perpendicular to the sliding direction. The cracks removed in the form of small broken sheet and flakes were suggestive of delamination wear [8,9,13].

At higher speed (2.60 m/s) and higher load (58.86 N) the oxidation and delamination were completely converted into adhesion wear. The higher load and velocity pin materials became soft; there were also a sign of plastic deformation and smearing. The region of the dominance of the wear mechanism depends on the normal load and sliding speed. Abrasion, oxidation, and delamination wear mechanisms were generally dominant in the lower sliding velocity and lower load region while adhesion and thermal softening/melting were dominant in the higher sliding velocity and loads. Figure 9 shows the energy-dispersive X-ray spectroscopy of some worn surfaces. The oxidation predominantly occurred on all rubbing surfaces during testing. The amount of oxidation varied with normal load and sliding velocity.

## 4. Conclusions

The sliding wear behaviour of the AZ91 alloy and its composite with varying content of SiC_p_ were studied. The following conclusions were drawn based on experiments performed:The microhardness of the SiC reinforced AZ91 composites was more than the unreinforced AZ91 alloy due to the presence of a hard particle in a soft matrix.The wear rate and coefficient of friction of the SiC reinforced composites was lower than the unreinforced composites. This is due to the higher hardness and grain refinement in SiC_p_ in composites.The wear rate increased with an increase in normal loads due to the increase in plastic deformation at higher load.The friction coefficient decreased with an increase in normal load and sliding distance due to a decrease in the mating surfaces’ asperities.The wear rate at velocity 1.39 m/s was higher than that at velocity 2.60 m/s in alloy and composites, and the effect of differences in velocity on wear rate decreased with the increase in normal loads. This may be because the effect of load on wear rate is higher than velocity and increase in hardness on addition of SiC_p_.The wear behaviour at velocity 1.39 m/s was dominated by oxidation and delamination wear whereas at velocity 2.6 m/s wear behaviour was dominated by abrasion and adhesion wear due to the induction of high energy at higher velocity and load.It was observed from SEM that the plastic deformation and smearing occurred at higher load and sliding velocity due to severe plastic deformation on the surface.

Thus, it may be stated that the magnesium alloy composites have potential applications in automobile industries, such as brakes and engine cylinders. From this study, it can be concluded that the addition of SiC in AZ91 can improve its wear behaviour and is suitable for applications such as transportation and defence industries. The fabricated AZ91 alloy-based metal matrix composites in this work are expected to be considered as lighter materials offering some benefits for tribological applications.

## Figures and Tables

**Figure 1 materials-14-00990-f001:**
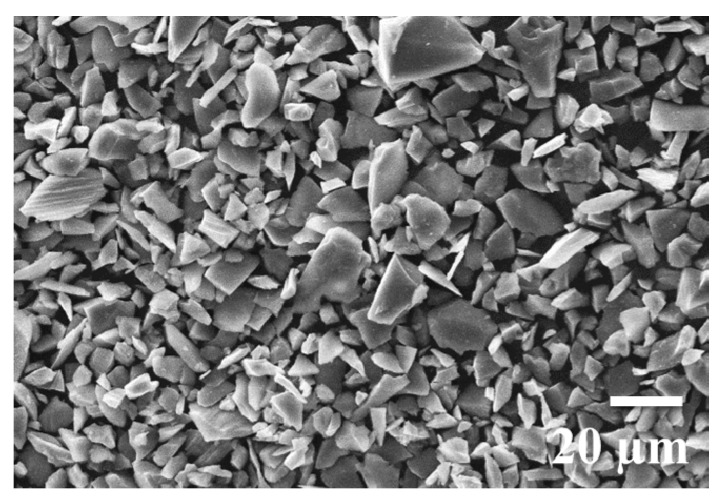
SEM micrograph of SiC_p_ reinforcement.

**Figure 2 materials-14-00990-f002:**
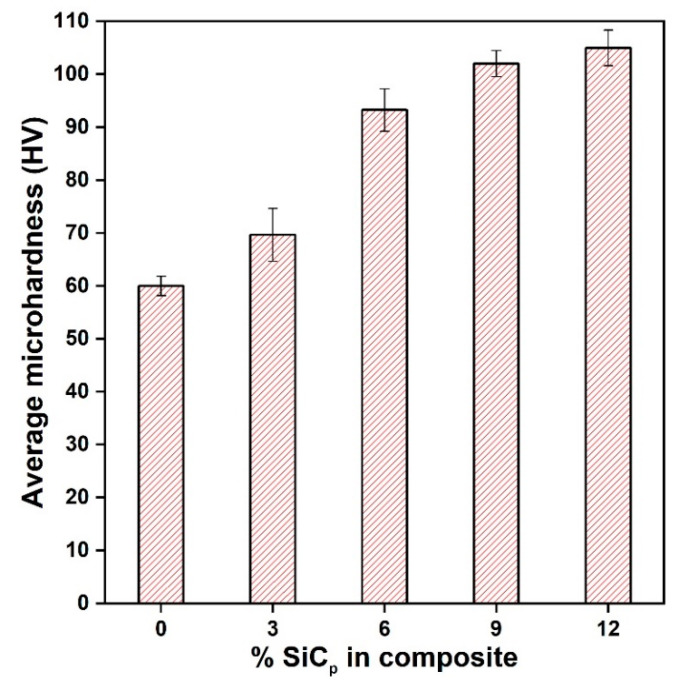
Microhardness of different composites with varying content of SiC.

**Figure 3 materials-14-00990-f003:**
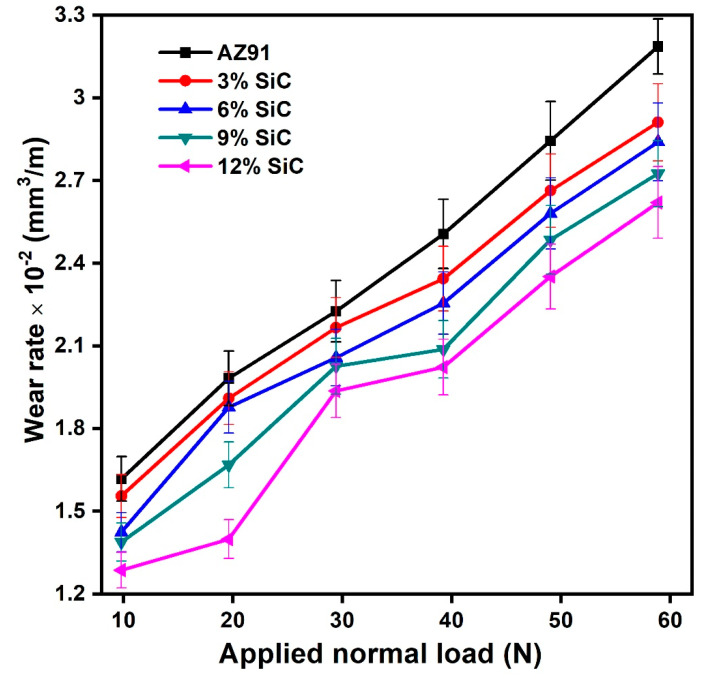
The wear rate variation with an applied normal load of AZ91 magnesium alloy and its composites.

**Figure 4 materials-14-00990-f004:**
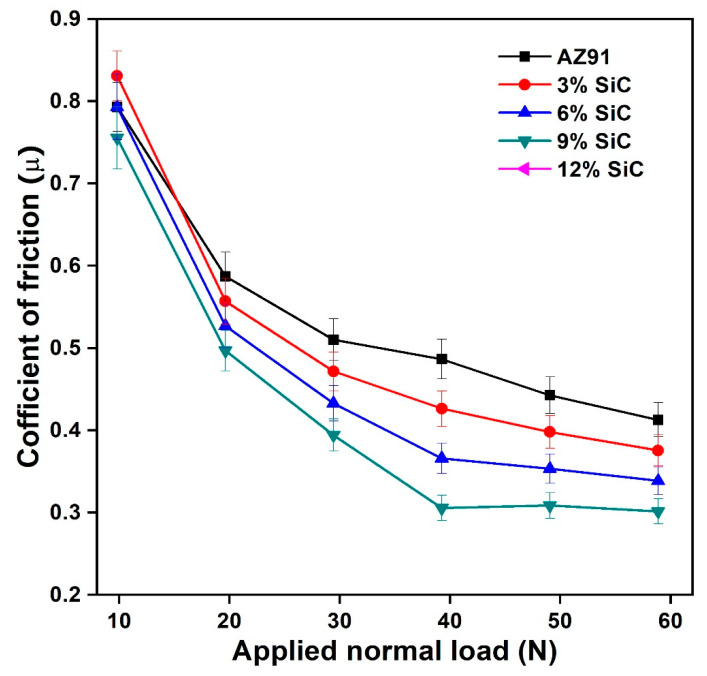
The variation of average coefficient of friction with an applied normal load of AZ91 magnesium alloy and its composite.

**Figure 5 materials-14-00990-f005:**
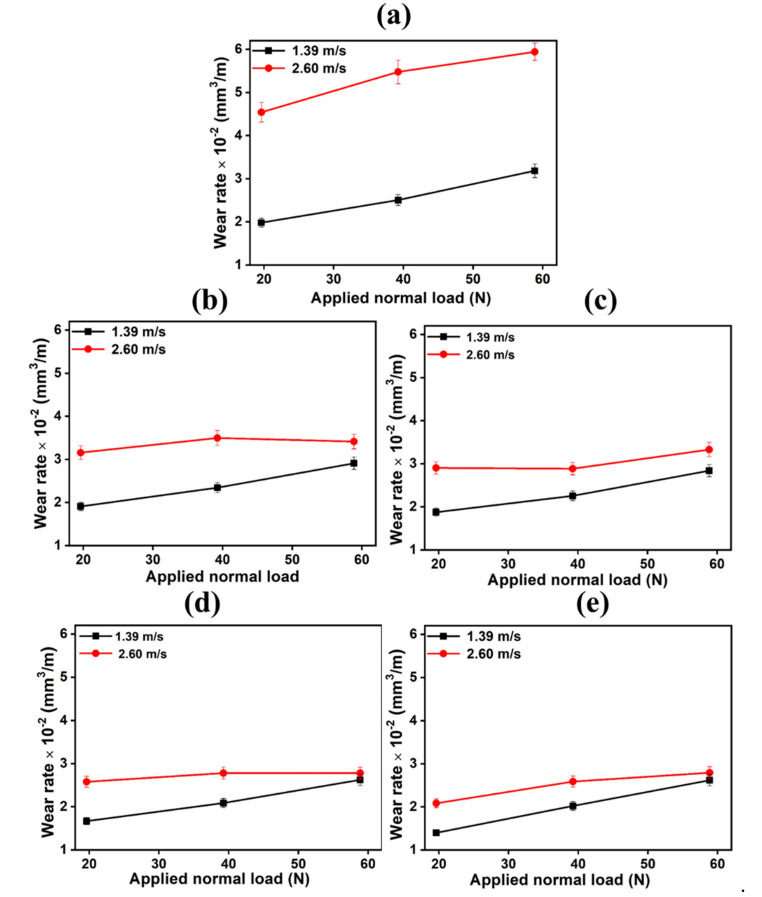
Variation of wear loss at different velocity and applied normal load for composites reinforced with SiC at (**a**) 0%; (**b**) 3%; (**c**) 6%; (**d**) 9%; (**e**) 12%.

**Figure 6 materials-14-00990-f006:**
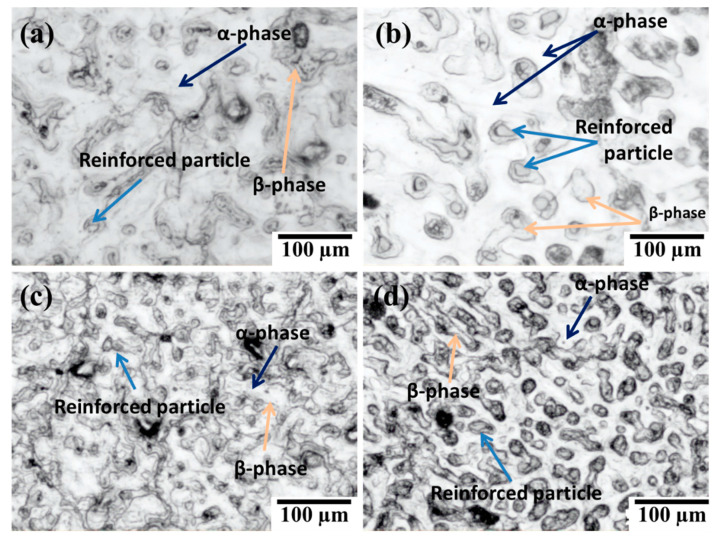
Optical micrographs for composites reinforced with SiC at (**a**) 3%; (**b**) 6%; (**c**) 9%; (**d**) 12%.

**Figure 7 materials-14-00990-f007:**
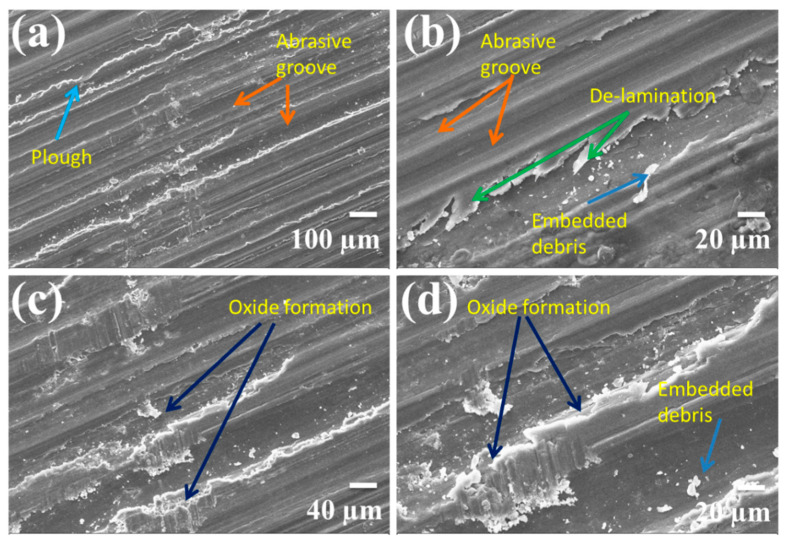
SEM micrographs of the worn surfaces of AZ91 magnesium alloy: (**a**) 19.24 N, 1.39 m/s; (**b**) 39.62 N, 2.6 m/s; (**c**) 19.24 N, 1.39 m/s; (**d**) 39.24 N, 2.6 m/s.

**Figure 8 materials-14-00990-f008:**
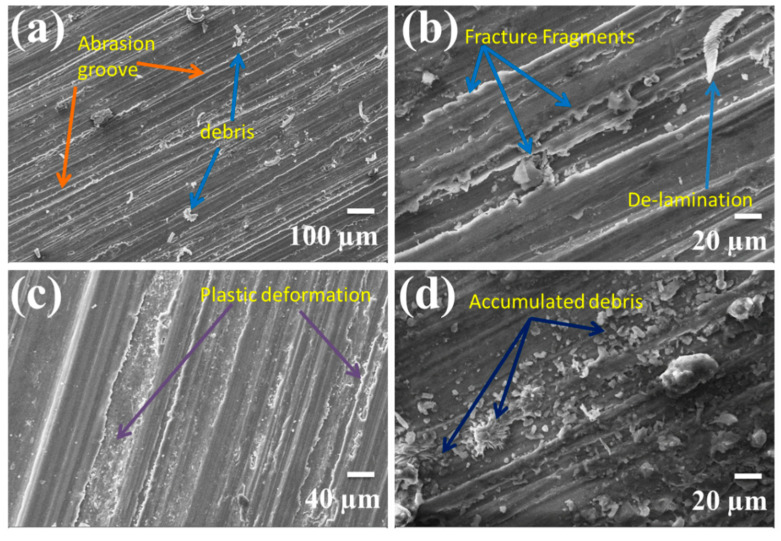
SEM micrographs of the worn surfaces of SiC reinforced AZ91 magnesium alloy composite: (**a**) 19.62 N, 2.6 m/s 3% SiC; (**b**) 39.24 N, 1.3 m/s and 6% SiC; (**c**) 58.86, 2.6 m/s 9% SiC; (**d**) 39.24 N, 2.6 m/s 12% SiC.

**Figure 9 materials-14-00990-f009:**
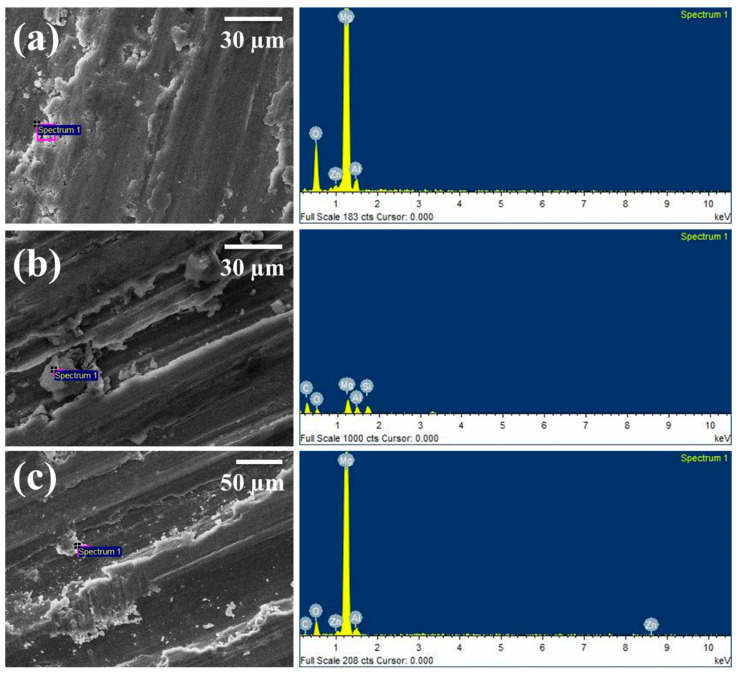
Energy-dispersive X-ray spectroscopy of worn surfaces composites: (**a**) 19.62 N, 1.30 m/s and 0% SiC; (**b**) 39.24 N, 1.3 m/s and 6% SiC; (**c**) 58.86, 2.6 m/s 9% SiC.

**Table 1 materials-14-00990-t001:** Parameters for the pin on disk dry sliding wear test.

Parameters	Value/Description
Pin Materials	AZ91 and its composite reinforced with SiC (3%, 6%, 9%, and 12%)
The Dimension of the Pin	30 mm length and 8 mm in diameter
Disk Material	Tool steel grade EN31
Track diameter (D)	83 mm for all experiments
Load Variation (N)	9.81 N, 19.62 N, 29.43 N, 39.24 N, 49.05 N, 58.86 N (for RPM-320) 19.62 N, 39.24 N, 58.86 N (for RPM-600)
Sliding Time (t)	15 min
Speed Variation (n)	320 RPM and 600 RPM

**Table 2 materials-14-00990-t002:** Summary of wear mechanisms for different combinations of normal load, sliding speed and pin materials.

Load	Sliding Speed (m/s)	Pin Material	Wear Mechanism				
			Abrasion	Oxidation	Delamination	Adhesion	Softening/Melting
19.62	1.39	AZ91	*	**	**		
		AZ91+3% SiC	*	**	**		
		AZ91+6% SiC	*	**	**		
		AZ91+9% SiC	*	**	**		
		AZ91+12% SiC	*	**	**		
19.62	2.6	AZ91	*	*	**	*	
		AZ91+3% SiC	*	*	**	*	
		AZ91+6% SiC	*	*	**	*	
		AZ91+9% SiC	*	*	**	*	
		AZ91+12% SiC	*	*	**	*	
39.24	1.39	AZ91	**	**	*	**	
		AZ91+3% SiC	**	**	*	**	
		AZ91+6% SiC	**	**	*	**	
		AZ91+9% SiC	**	**	*	**	
		AZ91+12% SiC	**	**	*	**	
39.24	2.6	AZ91	**	*	*	**	
		AZ91+3% SiC	**	*	*	**	
		AZ91+6% SiC	**	*	*	**	
		AZ91+9% SiC	**	*	*	**	
		AZ91+12% SiC	**	*	*	**	
58.86	1.39	AZ91	*			***	
		AZ91+3% SiC	*			***	
		AZ91+6% SiC	*			***	
		AZ91+9% SiC	*			***	
		AZ91+12% SiC	*			***	*
58.86	2.6	AZ91	*			***	*
		AZ91+3% SiC	*			***	*
		AZ91+6% SiC	*			***	*
		AZ91+9% SiC	*			***	*
		AZ91+12% SiC	*			***	*

Predicted relative extent of each wear mechanism: * slight; ** moderate; *** heavy.

## Data Availability

The authors confirm that the data supporting the findings of this study are available within the article.

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
