# Peer review of "Effect of Variation of SiC Reinforcement on Wear Behaviour of AZ91 Alloy Composites"

_materials, 2021, doi:10.3390/ma14040990_

Round 1

Reviewer 1 Report

Authors present a study about the effect of SiC reinforcement on wear behavior of AZ91 alloy composites.

This research manuscript may shows interest to the audience of this journal. The abstract is concise but not sufficient; Tables and figures are appropriate but needs to be improved. Nevertheless, some observations, and changes are recommended. This manuscript can be considered for publication after addressing the following comments:

1.Some minor redaction mistakes have been detected, please take a moment to revise and correct them.

2. Abstract section needs to address some values regarding your main findings as well.

3. Conclusions section needs to address some values regarding your main findings as well.

4. Roughness is a factor of great influence on the frictional wear of surfaces, it is recommended to include surface roughness measurements prior to wear tests.

5. Hardness is a highly influential factor in frictional wear. For surface wear, it is recommended to include surface hardness measurements prior to wear tests.

6. The manufacturer and, if possible, the measurement uncertainty of the balance used for weighing the wear test items should be indicated.

7. It must be indicated how the density of the materials is obtained for the calculation of the worn volume.

8. Although ASTM G99 recommends evaluation by weight loss, it is also recommended by measuring wear groove dimensions. This is due to the fact that part of the worn material shows as adhesion and deformation. Therefore, it is recommended to complement the evaluation with wear groove measurements.

9. Formulas 1 and 2 do not provide relevant information to the manuscript, it is recommended to eliminate them.

10. In order to apply Archard's equation, the hardness of the test material needs to be evaluated.

11. The selection of the tribological pair must be justified.

12. Error bars should be added to the results presented in the figures/graphs.

13. It is recommended to use the same scale on the vertical axes on all graphs where the wear rate is represented.

14. Figure 2 is equivalent to figure 1. It is recommended to eliminate one of the two figures.

15. The results presented in Figure 4 should be described in more depth, as they can be confusing.

16. The purpose of the results presented in Figures 5 to 9 should be clearly justified.

17. To justify the formation of oxides in Figures 10 and 11, it is necessary to include compositional analysis such as EDS or similar.

18. The sample and area on which the Energy-dispersive X-ray spectroscopy of worn surfaces of Figure 12 has been performed must be indicated.

19. The conclusions only describe what is shown in the graphs, it is recommended to rewrite the conclusions providing conclusive hypotheses as to why the phenomena observed in the results occur.

Author Response

Response to Reviewer #1

The authors would like to thank all the reviewers and the editor for their critical comments and useful suggestions, based on which we have improved the manuscript. Comments from all the reviewers have been carefully addressed in the revised manuscript. The changes made in the revised manuscript are highlighted. The itemized responses to the comments of all the reviewers are provided.

Reviewer #1

Authors present a study about the effect of SiC reinforcement on wear behavior of AZ91 alloy composites.

This research manuscript may shows interest to the audience of this journal. The abstract is concise but not sufficient; Tables and figures are appropriate but needs to be improved. Nevertheless, some observations, and changes are recommended. This manuscript can be considered for publication after addressing the following comments:

Comment 1.Some minor redaction mistakes have been detected, please take a moment to revise and correct them.

Authors' Response: The redaction mistakes in the manuscript has been corrected in the revised manuscript. The corrections are highlighted.

Comment 2. Abstract section needs to address some values regarding your main findings as well.

Authors' Response: As per the suggestion of reviewer, the abstract has been revised and data of main findings has been added to the abstract in the revised manuscript. The additions are highlighted.

Comment 3. Conclusions section needs to address some values regarding your main findings as well.

Authors' Response: As per the suggestion of reviewer, the conclusion section has been revised and data of main findings has been added in the revised manuscript. The additions are highlighted.

Comment 4. Roughness is a factor of great influence on the frictional wear of surfaces, it is recommended to include surface roughness measurements prior to wear tests.

Authors' Response: The roughness data has been included in the manuscript. The additions are highlighted.

Comment 5. Hardness is a highly influential factor in frictional wear. For surface wear, it is recommended to include surface hardness measurements prior to wear tests.

Authors' Response: The hardness data has been included in Fig 2 in the manuscript. The additions are highlighted.

Comment 6. The manufacturer and, if possible, the measurement uncertainty of the balance used for weighing the wear test items should be indicated.

Authors' Response: The manufacturer and the measurement uncertainty of the balance used for weighing the wear test items has been added in the manuscript. The additions are highlighted.

Comment 7. It must be indicated how the density of the materials is obtained for the calculation of the worn volume.

Authors' Response: The procedure for density measurement has been added to the materials and methods sections in the manuscript.

Comment 8. Although ASTM G99 recommends evaluation by weight loss, it is also recommended by measuring wear groove dimensions. This is due to the fact that part of the worn material shows as adhesion and deformation. Therefore, it is recommended to complement the evaluation with wear groove measurements.

Authors' Response: Authors followed ASTM G99-17 and according to ASTM G99-17, load , speed, sliding distance, track diameter, pin-end diameter, were considered. The wear volume loss and coefficient of friction was evaluated for reporting the results. To examine the types of wear mechanism SEM and EDX were performed. Wear grooves were not measured at the time of experimentation and now not possible due to surface oxidation of worn sample.

Comment 9. Formulas 1 and 2 do not provide relevant information to the manuscript, it is recommended to eliminate them.

Authors' Response: As per the suggestion of reviewer, formulas 1 and 2 have been eliminated in the revised manuscript.

Comment 10. In order to apply Archard's equation, the hardness of the test material needs to be evaluated.

Authors' Response: The hardness data of disk material as well as pin materials has been included in  the revised manuscript. The additions are highlighted.

Comment 11. The selection of the tribological pair must be justified.

Authors' Response: The justification for selection of the tribological pair has been added in the material and method section of the manuscript. The additions are highlighted.

Comment 12. Error bars should be added to the results presented in the figures/graphs.

Authors' Response: As per the suggestion of reviewer, error bars have been added to the results presented in the all figures in which it is required.

Comment 13. It is recommended to use the same scale on the vertical axes on all graphs where the wear rate is represented.

Authors' Response: In the revised manuscript, authors have used the same scale on the vertical axes of all graphs where the wear rate is represented.

Comment 14. Figure 2 is equivalent to figure 1. It is recommended to eliminate one of the two figures.

Authors' Response: The figure 1 has been removed from the revised manuscript.

Comment 15. The results presented in Figure 4  should be described in more depth, as they can be confusing.

Authors' Response: The figure 4 of  original manuscript has been removed as it was representing the same information similar to Figure 4 in revised manuscript.

Comment 16. The purpose of the results presented in Figures 5 to 9 should be clearly justified.

Authors' Response: The results presented in Figures 5 to 9 (now figure 5) has been described in more depth in the revised manuscript. The additions are highlighted.

Comment 17. To justify the formation of oxides in Figures 10 and 11, it is necessary to include compositional analysis such as EDS or similar.

Authors' Response: To justify the formation of oxides in Figures 10 and 11 (original manuscript), EDS has been included in Figure 9 of the revised manuscript.

Comment 18. The sample and area on which the Energy-dispersive X-ray spectroscopy of worn surfaces of Figure 12 has been performed must be indicated.

Authors' Response: The sample and area on which the EDX of worn surfaces of Figure 12 (original manuscript) has been performed is now presented in Figure 9.

Comment 19. The conclusions only describe what is shown in the graphs, it is recommended to rewrite the conclusions providing conclusive hypotheses as to why the phenomena observed in the results occur.

Authors' Response: The conclusion section has been revised and data of main findings has been added in the revised manuscript. The additions are highlighted.

Reviewer 2 Report

In the article “Effect of SiC Reinforcement on Wear Behaviour of AZ91 alloy Composites” a detailed comparative study of the sliding wear behaviors of the matrix alloy and its composite with different percentage of SiC particulates are presented. The Authors tried to reveal the different wear mechanism of the alloy and composites depend on the different combination of sliding velocity and applied normal load. The theme of the article is very interesting and relevant. But, a few comments should be providing.

  1. What method was used to obtain the composites containing different percentages of SiC particles?
  2. What were the sizes of the SiC particles?
  3. Research methods are not described in the manuscript. What equipment was used to study the obtained samples?
  4. The SEM images of composites are not presented in the manuscript. There is no information about their structure and about the distribution of the particles.
  5. The Authors wrote that oxidation mechanism took place during wear. Figures 10 (c, d) and 11 (c, d) show the zones of the oxide formation. But, only one EDX spectrum is shown in the work. What sample was it made for? It is not enough for proof. It would be great to demonstrate the EDX maps of the elements distribution.
  6. In the lines 144–145, the Authors wrote, “The combined effect of the different wear mechanism is involved in the estimation of wear loss”. But, in this case, it is not clear what the authors mean, because the five wear mechanisms are described below in the lines 221–224.
  7. In conclusion, there is no information about the wear mechanisms that were found in samples of the alloy and composites.

Author Response

Response to Reviewer #2

The authors would like to thank all the reviewers and the editor for their critical comments and useful suggestions, based on which we have improved the manuscript. Comments from all the reviewers have been carefully addressed in the revised manuscript. The changes made in the revised manuscript are highlighted. The itemized responses to the comments of all the reviewers are provided.

Reviewer #2

In the article “Effect of SiC Reinforcement on Wear Behaviour of AZ91 alloy Composites” a detailed comparative study of the sliding wear behaviors of the matrix alloy and its composite with different percentage of SiC particulates are presented. The Authors tried to reveal the different wear mechanism of the alloy and composites depend on the different combination of sliding velocity and applied normal load. The theme of the article is very interesting and relevant. But, a few comments should be providing.

Comment 1: What method was used to obtain the composites containing different percentages of SiC particles?

Authors' Response: The stir casting method was used to obtain the composites containing different SiC particles' percentages.

Comment 2: What were the sizes of the SiC particles?

Authors' Response: The average size of the SiC particulates was 15 micron.  For better clarity, an SEM image of SiC particulates has been added in the revised manuscript.

Comment 3: Research methods are not described in the manuscript. What equipment was used to study the obtained samples?

Authors' Response: The research methods have been added in the revised manuscript. The additions are highlighted.

Comment 4: The SEM images of composites are not presented in the manuscript. There is no information about their structure and about the distribution of the particles.

Authors' Response: As per the suggestion of the reviewer, the optical micrograph are presented in the Figure 6 instead of SEM images of the composites  in the revised manuscripts to show the distribution of the reinforced particles.

Comment 5: The Authors wrote that oxidation mechanism took place during wear. Figures 10 (c, d) and 11 (c, d) show the zones of the oxide formation. But, only one EDX spectrum is shown in the work. What sample was it made for? It is not enough for proof. It would be great to demonstrate the EDX maps of the element distribution.

Authors' Response: The EDX spectrum of other sample are also added in Figure 9 of the revised manuscript. Elemental mapping of the worn sample were not taken by the authors at time of wear and it not possible to evaluate right now because of huge oxide layer has been deposited on worn sample.

Comment 6: In the lines 144–145, the Authors wrote, “The combined effect of the different wear mechanism is involved in the estimation of wear loss”. But, in this case, it is not clear what the authors mean, because the five wear mechanisms are described below in the lines 221–224.

Authors' Response: The exact quantification of each wear mechanism is not possible. It is also not possible to excludes any wear mechanism completely out these five. The wear mechanisms vary from one mechanism to another depending on the load, velocity, and hardness of the mating parts. For clarity of above statement some additional explanation has been added in the manuscript. The additions are highlighted.

Comment 7: In conclusion, there is no information about the wear mechanisms that were found in samples of the alloy and composites.

Authors' Response: The conclusion section has been revised, and data of the main findings have been added in the revised manuscript. The additions are highlighted.

Round 2

Reviewer 1 Report

All the suggested modifications have been considered, and the manuscript have been improved in content and style. I reccomended this manuscript for publication after the modifications.

Please indicate the roughness parameter in L137. Ra? Rz?

Author Response

Response to Reviewer #1

The authors would like to thank the Reviewer and the Editor for their critical comments and useful suggestions on the basis of which we have refined the manuscript. The reviewer's comment is detailed in the revised manuscript. The changes made in the revised manuscript are highlighted. A detailed response to the reviewer's comment was provided.

Reviewer #1

All the suggested modifications have been considered, and the manuscript have been improved in content and style. I recommend this manuscript for publication after the modifications.

Comment 1. Please indicate the roughness parameter in L137. Ra? Rz?

Authors' Response: The roughness is average roughness and it is now corrected in the revised manuscript. The correction is highlighted.

Reviewer 2 Report

The Authors have made significant changes, the manuscript has been improved. I am very pleased with the presented work.

Author Response

Response to Reviewer #2

The authors would like to thank the Reviewer and the Editor for critical comments and useful suggestions based on which the manuscript was improved.

Reviewer #2

The Authors have made significant changes, the manuscript has been improved. I am very pleased with the presented work.

Authors' Response: The authors would like to thank the Reviewer once again for giving advice and accepting the changes made.
